# Anemia among pregnant women in Cambodia: A descriptive analysis of temporal and geospatial trends and logistic regression-based examination of factors associated with anemia in pregnant women

**Samnang Um**[1]*, **Heng Sopheab**[1], **An Yom**[1], **Jonathan A. Muir**[2‡]

**1** The National Institute of Public Health, Tuol Kork District, Phnom Penh, Cambodia, **2** The Global Health Institute, Emory University, Atlanta, Georgia, United States of America

‡ JAM also contributed equally to this work.

* umsamnang56@gmail.com

**Data Availability Statement:** Our study used 2005, 2010, and 2014 Cambodia Demographic and Health Survey (CDHS) datasets. The DHS data are

## Abstract

Anemia is a major public health problem for thirty-two million pregnant women worldwide. Anemia during pregnancy is a leading cause of child low birth weight, preterm birth, and perinatal/neonatal mortality. Pregnant women are at higher risk of anemia due to micronutrient deficiencies, hemoglobinopathies, infections, socio-demographic and behavioral factors. This study aimed to: 1) assess temporal and geospatial trends of anemia in Cambodia and 2) identify factors associated with anemia among pregnant women aged 15–49 years old in Cambodia. We analyzed data from the Cambodia Demographic and Health Survey (CDHS) for 2005, 2010, and 2014. Data were pooled across the three survey years for all pregnant women aged 15–49 years. Survey weights were applied to account for the complex survey design of the CDHS. Descriptive statistics were estimated for key sociodemographic characteristics of the study population. We used logistic regressions to assess factors associated with anemia among pregnant women aged 15–49 years old. Anemia in pregnant women aged 15–49 in Cambodia decreased from 56% in 2005 to 53% in 2014. With the highest in Preah Vihear and Stung Treng provinces (74.3%), in Kratie province (73%), and in Prey Veng (65.4%) in 2005, 2010, and 2014 respectively. Compared to pregnant women from the wealthiest households, women from poorest households were more likely to have anemia (AOR = 2.8; 95% CI: 1.6–4.9). Pregnant women from coastal regions were almost twice as likely of having anemia (AOR = 1.9; 95% CI: 1.2–3.0). Pregnant women were more likely anemic if they were in their 2nd trimester (AOR = 2.6; 95% CI: 1.9–3.6) or 3rd trimester (AOR = 1.6 95% CI: 1.1–2.3). Anemia remains highly prevalent among pregnant women in Cambodia. Public health interventions and policies to alleviate anemia should be prioritized and shaped to address these factors.

publicly available from the website: https://dhsprogram.com.

**Funding:** The author(s) received no specific funding for this work.

**Competing interests:** The authors have declared that no competing interests exist.

**Abbreviations:** ANC, Antenatal Care; AOR, Adjusted Odds Ratio; CDHS, Cambodian Demographic Health Survey; EA, Enumeration areas; Hb, Hemoglobin; WHO, World Health Organization.

## Introduction

Anemia during pregnancy occurs in both developed and developing countries and is associated with abnormal outcomes in pregnancy; for example, greater risk for low birth weight, preterm birth, and perinatal/neonatal mortality [1]. According to the World Health Organization (WHO), anemia is defined as hemoglobin (Hb) levels less than 11.0 g/dL in pregnant women [2]. In 2019, it was estimated that anemia affected approximately 37% (32 million) of pregnant women worldwide and as many as half of all pregnant women in low-income and middle-income countries were diagnosed with anemia [3]. In Southeast Asia countries, 48% of pregnant women were anemic [4]. These patterns are reflected in Cambodia where the prevalence of anemia is higher among pregnant women compared with non-pregnant women (i.e., 53.2% compared to 45%) [5].

Anemia in pregnant women has severe consequences on health as well as social and economic development [6]. contributing to low physical activity and increases maternal morbidity and mortality, especially among those with severe anemia [1] and those living in developing countries [7]. The WHO estimates that 20% of maternal deaths are attributable to anemia [2]. In Southeast Asia, half of all maternal deaths were due to anemia in 2016 [8]. In Cambodia, anemia is affected 51.1% of pregnant women aged 15–49 years old in 2019 [9]. Despite significant economic development in Cambodia over the past two decades, the prevalence of anemia in women has not substantially declined.

Pregnant women with anemia are twice as likely to die during or shortly after pregnancy compared to those without anemia [8]. In many countries, anemia varies by socioeconomic factors such as education, household wealth status, occupation, and residence [10]. Pregnant women aged 30 and older are more likely to develop anemia than younger aged pregnant women [11–13]. Risk of anemia was greater among women living in the lowest wealth quintile, with limited education [14], and rural area [12, 13]. Similarly, repeated childbearing, inadequate water hygiene and sanitation status, and parasitic infection increases the risk of anemia in pregnant women [15]. Factors associated with anemia during pregnancy in developing countries include nutritional deficiencies of iron, vitamin B12, folate, and parasitic diseases (for example, malaria) [6, 16, 17]. Compared to women who initiated antenatal care (ANC) in the first trimester, the odds of having anemia in pregnancy were significantly higher among pregnant women who initiated ANC in the second trimester (AOR = 2.71) and third trimester (AOR = 5.01) [18].

The prevalence of anemia in Cambodia has slowly decreased in both pregnant and non-pregnant women. To further reduce anemia in Cambodia, improved understanding of its geographical distribution and associated risk factors is required. Enhanced understanding will help identify populations at greater risk for anemia and prioritize geographic areas for targeted interventions. To our knowledge, there are no published peer-reviewed studies that assessed social and demographic factors associated with anemia among pregnant women in Cambodia over time. One prior study on anemia patterns and risk factors utilized data from the CDHS 2014 [14]. This study included all women of reproductive age 15–49 years and pooled DHS data from seven South and Southeast Asian countries. An additional study aimed to describe trends in nutritional status of women of reproductive age in Cambodia over four nationally representative Demographic Health Surveys (2000, 2005, 2010, and 2014) and to assess factors associated with nutritional disparities [19]. Finally, a research abstract published in association with a research conference in Cambodia referenced using CDHS 2014; this was a preliminary version of the present study [20]. Given the paucity of scholarship addressing this health concern in Cambodia, in this study we aimed to:

1. Assess temporal and geospatial trends of anemia among pregnant women in Cambodia

2. Identify factors associated with anemia among pregnant women in Cambodia

Understanding these factors may further support policy development and programs with more effective strategies and interventions in Cambodia to reduce the prevalence of anemia among pregnant women and associated health risks such as maternal mortality.

## Materials and methods

### Data

To analyze trends and factors associated with anemia in pregnant women in Cambodia, we used existing women's data from the 2005, 2010, and 2014 CDHS. The CDHS is a nationally representative population-based household survey that is regularly conducted roughly every 5 years. The survey typically uses two-stage stratified cluster sampling to collect samples from all provinces that are divided into sampling domains. In the first stage, clusters, or enumeration areas (EAs) that represent the entire country are randomly selected from the sampling frame using probability proportional to cluster size (PPS). The second stage then involves the systematic sampling of households listed in each cluster or EA. Interviews were then conducted with women aged 15–49 in selected households. Using the women's survey questionnaire, variables collected by the CDHS include births to women aged 15–49, sociodemographic characteristics, household assets that are used to calculate a household wealth index, health-related indicators and nutritional status, number of ANC visits and other pregnancy/delivery indicators for past births, and involvement in household decision making using questionnaires. It further includes height and weight measurements of women and children, hemoglobin level, malaria test, and vitamin A level [5]. We restricted our sample to pregnant women with recorded hemoglobin level; which resulted in a total sample size of 1,629 (1,567 weighted) pregnant women aged 15–49 (Fig 1).

### Measurements

The outcome variable used in the study was anemia, which in the CDHS was based on hemoglobin levels that were measured by collecting capillary blood from a finger prick with the HemoCue 201+analyzer [19]. The variable was originally coded as a categorical variable with not anemic, mild anemia, moderate anemia, and severe anemia. For the purpose of this study, we recoded the original variable into a dichotomous variable anemia, with mild, moderate, and severe anemia coded as Anemic = 1 and non-Anemic = 0.

Right-hand-sided variables included women and household characteristics. Women's age was categorized into an ordinal level variable with 15–20 years = 1, 21–31 years = 2 (reference category), and 32–49 years = 3. Women's marital status was coded into a dichotomous variable with Married = 1, Not Married = 0 (includes divorce, single, widowed, and separated). Women's education was coded as an ordinal level variable with Higher = 1 (reference category), Secondary = 2, Primary = 3, No Education = 4. Women's occupation was coded as a categorical variable with Professional = 1 (reference category; included clerical, technical, managerial, professional, sales, and services), Agriculture/Manual = 2 (included skilled and unskilled manual work), Not Working = 3 (1 observation had a missing value). Household level wealth was measured using a wealth index that was coded as an ordinal level variable with richest = 1 (reference category), richer = 2, middle = 3, poorer = 4, and poorest = 5 (we opted to use the wealth index that was provided by each respective survey year of the CDHS as opposed to calculating a wealth index across the pooled data. This decision was based on prior research that found

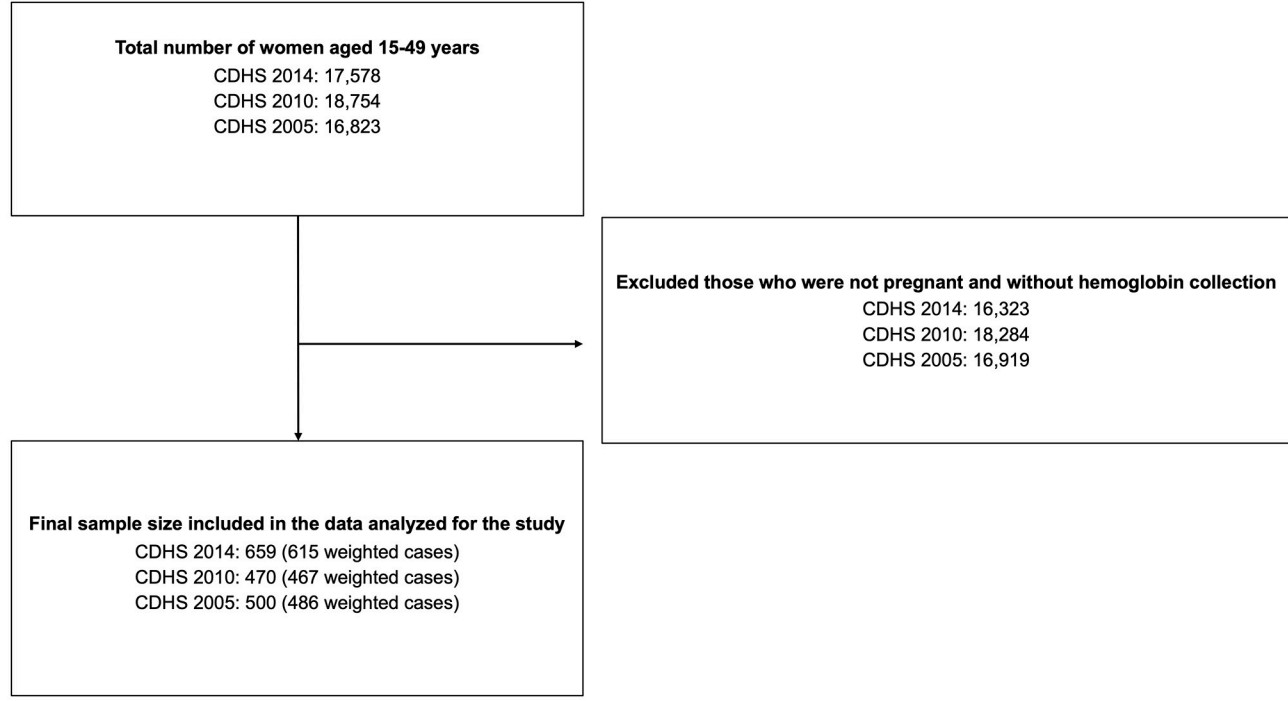

**Fig 1. Selection process and the final sample size from the 2005, 2010, and 2014 CDHS.**

wealth status designation was comparable across the CDHS provided indices and an index based on data pooled across years) [20, 21].

Pregnancy duration (trimester) was coded as an ordinal level variable with $1^{st} = 1$ (*reference category*), $2^{nd} = 2$, and $3^{rd} = 3$. BMI was coded as an ordinal level variable with Overweight or Obese = 1 (reference category), Normal = 2, and Underweight = 3. Prior births was coded as an ordinal level variable with None = 1 (reference category), 1–2 Births = 2, and 3+ Births = 3 (3 observations had a missing value). Tobacco use was coded as a dichotomous variable with No Tobacco Use = 0 and Tobacco Use = 1 (possible types of tobacco use included cigarettes, cigars, and chewing tobacco; 2 observations had a missing value).). Healthcare barriers was coded as a dichotomous variable with No Barriers = 0 and 1 or more barriers = 1 (possible barriers reported included distance, money, and waiting time; 1 observation had a missing value).

Residence was a dichotomous variable representing urban vs. rural place of residence with Rural = 1, Urban = 0. Phnom Penh was a dichotomous variable representing residence in the provincial area surrounding Phnom Penh vs. other regions of residence with Other Regions = 1, Phnom Penh = 0. Cambodia's domains/provinces were regrouped for analytic purposes into a categorical variable with 4 geographical regions that were coded as Plains = 1 (reference category), Tonle Sap = 2, Costal/sea = 3, and Mountains = 4. Health facility visit and aborted pregnancy were coded as dichotomous variables representing whether the respondent had visited a health facility in the past year or had ever terminated a pregnancy; results are not presented due to lack of statistical significance in unadjusted analyses.

## Analytic strategy

Data cleaning and analyses were performed using STATA version 16 (Stata Corp 2019, College Station, TX) [22]. Additional visualization was performed using R version 4.2.0 (code available

upon request); the complex survey design of the CDHS. which typically uses two-stage strati-fied cluster sampling, was accounted for using the "survey" package that enables specification of variables representing the enumeration areas, strata, and survey weights associated with the complex survey design [23, 24]. Sample weights and complex survey design were accounted for in both descriptive and logistic regression analyses; accounting for the complex survey design facilitates generating nationally representative results from these analyses. To convey temporal and geographic trends in anemia prevalence, weighted estimates of overall and region-specific temporal trends in anemia prevalence were visualized using the ggplot2 pack-age in R [25]. In addition, a temporal series of maps illustrating provincial variation in the prevalence of anemia were also prepared by our team using ArcGIS software version 10.8 [26]. The underlying shapefiles of Cambodia provinces were obtained from the publicly available Spatial Data Repository associated with the DHS website at [https://spatialdata.dhsprogram.com/boundaries/#view=table&countryId=KH] [27]. Data quality checks and minor data cleaning followed standard procedures in preparation for statistical analysis [28].

Cross tabulation chi-square tests were used to describe and provide preliminary assessment of associations between independent variables of interest including maternal demographic, household characteristics, geographical regions, and health-related) and anemia status. Simple logistic regression models and several pathway specific models were examined prior to fitting the final model (results from the pathway specific models are available upon request). Vari-ables were selected for inclusion in the final multiple logistic regression if their inclusion had theoretical justification [12]. Correlation among independent variables was assessed using a combination of cross tabulation chi-square tests and Cramer's V scores; all results suggested minimal to moderate correlation between independent variables (all Crammer's v scores were lower than .5, most were lower than .3). Further consideration of whether to include a combi-nation of variables was based on broader evaluations of model fit, which we assessed using Pseudo-$R^2$, Deviance, AIC, and BIC scores. The Pseudo-$R^2$ value was the largest and the Devi-ance, AIC, BIC scores were the smallest for the fully adjusted model, indicating that this model had superior fit compared to other models.

Simple logistic regression was used to analyze the magnitude effect of associations between anemia and maternal bio-demographic and household characteristics, geographical regions, and health-related factors. Results are reported as Odds Ratios (OR) with 95% confidence intervals (CI). Multiple logistics regression was then used to assess independent factors associ-ated with anemia after adjusting for other potential confounding factors in the model. Results from the final adjusted model are reported as adjusted odds ratios (AOR) with 95% confidence intervals and corresponding p-values. Results from the final multiple logistic regression model were considered as statistically significant based on a p-value less than 0.05 and 95% confi-dence intervals. Evaluations of effect modification were not statistically significant. Observa-tions with missing values were omitted from the regression analyses. All results were visualized as forest plots that were generated using the forest model package in R.

This study was approved by the National Ethical Committee for Health Research (Ref: 225 NECHR) Cambodia and the Institutional Review Board (IRB) of ICF in Rockville, Maryland, USA. The CDHS data are publicly accessible and were made available to us upon request to the DHS Program website at (https://dhsprogram.com/data/available-datasets.cfm) [29]. Written consents were obtained from all participants before the interview with participants in the CDHS.

## Results

A little over 60% of pregnant women included in the analysis were aged 21–30 years, and almost 98% were married (Table 1). Approximately 17% had no schooling, 52% had a primary

**Table 1. Descriptive characteristics of anemia prevalence among study population (n = 1,567).**

| Variables | 2005 | | | | | 2010 | | | | | 2014 | | | | | 2005–2014 | | | | |
|---|---|---|---|---|---|---|---|---|---|---|---|---|---|---|---|---|---|---|---|---|
| | Anemic | | Not Anemic | | p-value | Anemic | | Not Anemic | | p-value | Anemic | | Not Anemic | | p-value | Anemic | | Not Anemic | | p-value |
| | f | % | f | % | | f | % | f | % | | f | % | f | % | | f | % | f | % | |
| **Age (Years)** | | | | | | | | | | | | | | | | | | | | |
| 15–20 | 54 | 57.5 | 40 | 42.5 | 0.159 | 52 | 63.6 | 30 | 36.4 | 0.003 | 76 | 58.8 | 53 | 41.2 | 0.386 | 182 | 59.7 | 123 | 40.3 | **0.002** |
| 21–31 | 134 | 51.4 | 127 | 48.6 | | 150 | 47.2 | 169 | 52.8 | | 189 | 51.0 | 182 | 49.0 | | 474 | 49.8 | 477 | 50.2 | |
| 32–49 | 84 | 63.9 | 47 | 36.1 | | 44 | 67.6 | 21 | 32.4 | | 62 | 54.1 | 53 | 45.9 | | 190 | 61.1 | 121 | 38.9 | |
| **Marital Status** | | | | | | | | | | | | | | | | | | | | |
| Married | 269 | 56.2 | 210 | 43.8 | 0.270 | 237 | 52.4 | 215 | 47.6 | 0.215 | 318 | 52.8 | 284 | 47.2 | | 824 | 64.3 | 709 | 35.7 | **0.232** |
| Not Married | 2 | 36.4 | 4 | 63.6 | | 10 | 69.5 | 4 | 30.5 | | 10 | 72.0 | 4 | 28.0 | | 22 | 53.8 | 12 | 46.2 | |
| **Education** | | | | | | | | | | | | | | | | | | | | |
| Higher | 5 | 74.9 | 2 | 25.1 | 0.633 | 1 | 13.6 | 6 | 86.4 | 0.107 | 9 | 33.7 | 17 | 66.3 | 0.025 | 15 | 36.7 | 25 | 63.3 | 0.025 |
| Secondary School | 48 | 53.4 | 42 | 46.6 | | 57 | 47.4 | 63 | 52.6 | | 116 | 48.5 | 123 | 51.5 | | 221 | 49.2 | 228 | 50.8 | |
| Primary School | 148 | 54.2 | 125 | 45.8 | | 143 | 54.6 | 119 | 45.4 | | 170 | 59.5 | 116 | 40.5 | | 461 | 56.2 | 359 | 43.8 | |
| No Education | 71 | 61.0 | 45 | 39.0 | | 46 | 59.3 | 32 | 40.7 | | 33 | 50.8 | 32 | 49.2 | | 150 | 58.0 | 109 | 42.0 | |
| **Occupation** | | | | | | | | | | | | | | | | | | | | |
| Professional | 35 | 43.9 | 44 | 56.1 | 0.045 | 43 | 48.6 | 46 | 51.4 | 0.504 | 46 | 39.9 | 70 | 60.1 | 0.021 | 124 | 43.7 | 160 | 56.3 | **0.010** |
| Agricultural/ Manual Labor | 235 | 58.0 | 170 | 42.0 | | 157 | 52.5 | 142 | 47.5 | | 185 | 59.1 | 128 | 40.9 | | 577 | 56.7 | 440 | 43.3 | |
| Not Working | 2 | 100.0 | 0 | 0.0 | | 46 | 59.3 | 32 | 40.7 | | 96 | 51.7 | 90 | 48.3 | | 145 | 54.4 | 122 | 45.6 | |
| **Wealth Index** | | | | | | | | | | | | | | | | | | | | |
| Richest | 27 | 40.2 | 40 | 59.8 | 0.002 | 25 | 34.8 | 47 | 65.2 | 0.012 | 51 | 40.9 | 73 | 59.1 | 0.001 | 103 | 39.0 | 160 | 61.0 | <**0.001** |
| Richer | 43 | 43.4 | 57 | 56.6 | | 37 | 51.7 | 34 | 48.3 | | 64 | 51.8 | 60 | 48.2 | | 145 | 48.9 | 151 | 51.1 | |
| Middle | 51 | 57.0 | 39 | 43.0 | | 58 | 51.4 | 55 | 48.6 | | 40 | 39.3 | 62 | 60.7 | | 150 | 49.0 | 156 | 51.0 | |
| Poorer | 66 | 60.5 | 43 | 39.5 | | 50 | 52.7 | 45 | 47.3 | | 99 | 66.3 | 50 | 33.7 | | 215 | 60.9 | 138 | 39.1 | |
| Poorest | 84 | 70.4 | 35 | 29.6 | | 77 | 66.7 | 38 | 33.3 | | 73 | 63.4 | 42 | 36.6 | | 234 | 66.9 | 116 | 33.1 | |
| **Pregnancy Duration** | | | | | | | | | | | | | | | | | | | | |
| 1st Trimester | 53 | 45.5 | 64 | 54.5 | 0.028 | 53 | 37.3 | 89 | 62.7 | 0.001 | 75 | 43.8 | 96 | 56.2 | 0.003 | 182 | 42.1 | 250 | 57.9 | <**0.001** |
| 2nd Trimester | 112 | 65.0 | 60 | 35.0 | | 108 | 65.1 | 58 | 34.9 | | 150 | 63.7 | 86 | 36.3 | | 371 | 64.5 | 204 | 35.5 | |
| 3rd Trimester | 106 | 54.3 | 89 | 45.7 | | 85 | 54.2 | 72 | 45.8 | | 102 | 49.1 | 106 | 50.9 | | 294 | 52.4 | 267 | 47.6 | |
| **BMI** | | | | | | | | | | | | | | | | | | | | |
| Overweight or Obese | 67 | 45.1 | 71 | 54.9 | 0.042 | 72 | 43.7 | 66 | 56.3 | 0.524 | 27 | 47.8 | 29 | 52.2 | 0.081 | 72 | 45.6 | 86 | 54.4 | **0.005** |
| Normal | 182 | 61.4 | 114 | 38.6 | | 152 | 55.1 | 124 | 44.9 | | 184 | 58.3 | 132 | 41.7 | | 518 | 58.4 | 370 | 41.6 | |
| Underweight | 23 | 48.3 | 28 | 51.7 | | 23 | 52.0 | 29 | 48.0 | | 117 | 47.8 | 127 | 52.2 | | 255 | 49.1 | 265 | 50.9 | |
| **Prior Births** | | | | | | | | | | | | | | | | | | | | |
| None | 80 | 53.2 | 71 | 46.8 | 0.731 | 91 | 52.1 | 84 | 47.9 | 0.222 | 158 | 52.7 | 141 | 47.3 | 0.072 | 329 | 52.7 | 296 | 47.3 | **0.039** |
| 1–2 Births | 110 | 56.2 | 86 | 43.8 | | 105 | 49.5 | 107 | 50.5 | | 122 | 49.8 | 123 | 50.2 | | 337 | 5.1.6 | 316 | 48.4 | |
| 3+ Births | 82 | 58.7 | 58 | 41.3 | | 51 | 64.0 | 29 | 36.0 | | 48 | 67.2 | 23 | 32.8 | | 180 | 62.2 | 109 | 37.8 | |
| **Tobacco Use** | | | | | | | | | | | | | | | | | | | | |
| No | 250 | 54.6 | 208 | 45.4 | 0.015 | 234 | 52.6 | 211 | 47.4 | 0.564 | 320 | 52.7 | 288 | 47.3 | 0.021 | 805 | 53.2 | 707 | 46.8 | **0.027** |
| Yes | 22 | 78.6 | 6 | 21.4 | | 13 | 60.4 | 9 | 39.6 | | 7 | 100.0 | 0 | 0.0 | | 42 | 74.1 | 15 | 25.9 | |
| **Healthcare Barriers** | | | | | | | | | | | | | | | | | | | | |
| No Barriers | 43 | 48.4 | 45 | 51.6 | 0.215 | 50 | 44.3 | 63.0 | 55.7 | 0.073 | 81 | 49.7 | 82 | 50.3 | 0.342 | 174 | 47.7 | 191 | 52.3 | **0.019** |
| 1 or More Barriers | 226 | 57.3 | 169 | 42.7 | | 197 | 55.7 | 156.0 | 44.3 | | 246 | 54.5 | 205 | 45.5 | | 669 | 55.8 | 530 | 44.2 | |
| **Place of Residence** | | | | | | | | | | | | | | | | | | | | |
| Urban | 23 | 42.7 | 32 | 57.3 | 0.029 | 28 | 39.3 | 43 | 60.7 | 0.025 | 50 | 46.1 | 58 | 53.9 | 0.095 | 101 | 43.2 | 133 | 56.8 | <**0.001** |
| Rural | 248 | 57.7 | 182 | 42.3 | | 219 | 55.4 | 176 | 44.6 | | 278 | 54.7 | 230 | 45.3 | | 745 | 55.9 | 588 | 44.1 | |
| **Phnom Penh** | | | | | | | | | | | | | | | | | | | | |

*(Continued)*

**Table 1.** (Continued)

| Variables | 2005 | | | | | 2010 | | | | | 2014 | | | | | 2005–2014 | | | | |
|---|---|---|---|---|---|---|---|---|---|---|---|---|---|---|---|---|---|---|---|---|
| | Anemic | | Not Anemic | | p-value | Anemic | | Not Anemic | | p-value | Anemic | | Not Anemic | | p-value | Anemic | | Not Anemic | | p-value |
| | f | % | f | % | | f | % | f | % | | f | % | f | % | | f | % | f | % | |
| Phnom Penh | 38 | 41.9 | 27 | 58.1 | 0.047 | 34 | 39.3 | 22 | 60.7 | 0.110 | 49 | 47.5 | 54 | 52.5 | 0.31 | 98 | 43.8 | 126 | 56.2 | **0.010** |
| Other Regions | 176 | 58.1 | 245 | 41.9 | | 186 | 54.8 | 225 | 45.2 | | 278 | 54.4 | 233 | 45.6 | | 748 | 55.7 | 595 | 44.3 | |
| **Geographic Region** | | | | | | | | | | | | | | | | | | | | |
| Plain | 124 | 51.1 | 119 | 48.9 | 0.187 | 104 | 48.5 | 111 | 51.5 | 0.222 | 119 | 49.5 | 121 | 50.5 | 0.432 | 347 | 49.7 | 351 | 50.3 | **0.019** |
| Tonle Sap | 93 | 59.9 | 62 | 40.1 | | 86 | 53.6 | 74 | 46.4 | | 112 | 53.4 | 98 | 46.6 | | 290 | 55.4 | 234 | 44.6 | |
| Coastal | 18 | 66.0 | 9 | 34.0 | | 20 | 66.8 | 10 | 33.2 | | 26 | 62.7 | 16 | 37.3 | | 64 | 64.8 | 35 | 35.2 | |
| Plateau/ Mountain | 37 | 60.8 | 24 | 39.2 | | 37 | 60.0 | 25 | 40.0 | | 70 | 57.0 | 53 | 43.0 | | 144 | 58.7 | 102 | 41.3 | |
| **Survey Year** | | | | | | | | | | | | | | | | | | | | |
| 2005 | | | | | | | | | | | | | | | | 272 | 56.0 | 214 | 44.0 | 0.696 |
| 2010 | | | | | | | | | | | | | | | | 247 | 52.9 | 220 | 47.1 | |
| 2014 | | | | | | | | | | | | | | | | 327 | 53.2 | 288 | 46.8 | |

**Notes: Study sample size of 1,567. Survey weights applied to obtain weighted percentages. Plains:** Phnom Penh, Kampong Cham/Tbong Khmum, Kandal, Prey Veng, Svay Rieng, and Takeo; **Tonle Sap:** Banteay Meanchey, Kampong Chhnang, Kampong Thom, Pursat, Siem Reap, Battambang, Pailin, and Otdar Meanchey; **Coastal/sea:** Kampot, Kep, Preah Sihanouk, and Koh Kong; **Mountains:** Kampong Speu, Kratie, Preah Vihear, Stung Treng, Mondul Kiri, and Ratanak Kiri. Variables with missing values for fewer than 3 observations include: Occupation, BMI, Tobacco Use, and Healthcare Barriers.

education, 29% had a secondary education, and only 2.6% had a higher education. Roughly 45% were from poorest or poorer household wealth quintiles. Pregnancy duration was rather evenly distributed with 28% of pregnant women in their 1$^{st}$ trimester, 37% in their 2$^{nd}$ trimester, and 36% in their third trimester. Slightly over 50% of pregnant women had a normal BMI, with another 33% were either overweight or obese. About 80% reported having fewer than 3 prior births. Only about 6% of pregnant women reported using tobacco. A majority (85%) of them resided in rural areas (Table 1).

The overall prevalence of anemia among pregnant women decreased slightly from 56% in 2005 to 53% in 2014 (Fig 2). This decline varied across the four geographical regions with the highest prevalence of anemia in the coastal region in 2010 (66%). Plain region had the lowest prevalence of anemia across all survey years with prevalence levels in 2005 lower than the prevalence levels of other regions in 2014. Anemia was highest among pregnant women in Preah Vihear and Stung Treng provinces (74.3%), in Kratie province (73%), and in Prey Veng (65.4%) for 2005, 2010 and 2014 respectively (Fig 3). Pregnant women living in Phnom Penh had the lowest prevalence of anemia in 2005 and 2010 (i.e., 33.3% in 2005, 30.0% in 2010. In 2014, the lowest prevalence of anemia among pregnant women was observed in Takeo province (23.8%) (S1 Table).

Regarding socio-demographic factors associated with anemia in pregnancy in simple logistic regression analysis, pregnant women were more likely to have anemia if they were aged 15–20 (OR = 1.5; 95% CI: 1.1–2.0) or aged 31–49 (OR = 1.6; 95% CI: 1.1–2.2) compared to pregnant women aged 21–30 (Fig 4). Pregnant women with no formal education were more likely to have anemia compared to women with a higher education (2.4; 95% CI: 1.1–5.1) as were women with only a primary level education (OR = 2.2; 95% CI: 1.1–4.5). Women working in agricultural or manual labor fields were more likely to have anemia compared to women working in professional fields (OR = 1.7; 95% CI: 1.2–2.3). Juxtapose pregnant women from the richest wealth quintile, pregnant women with lower wealth were more likely to have anemia;

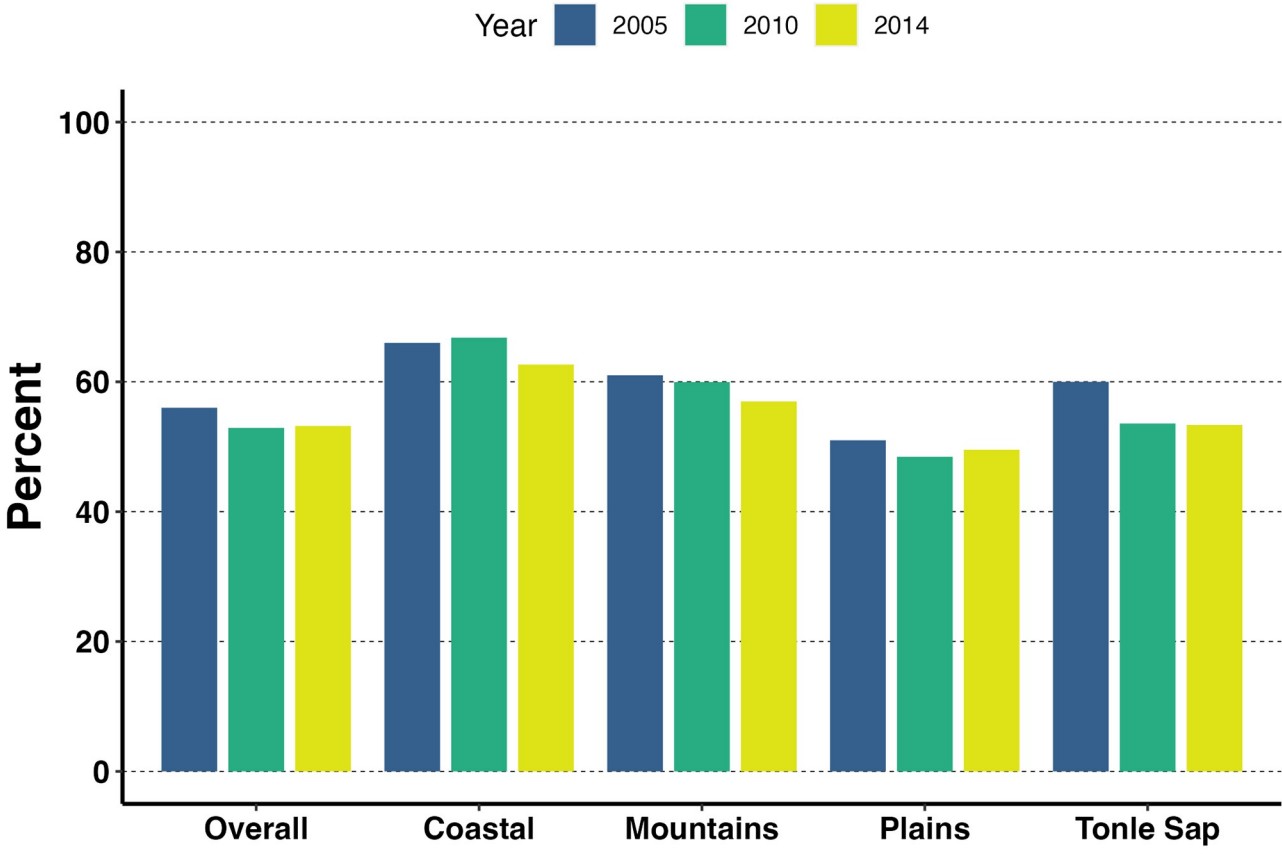

**Fig 2. Overall and regional trends of anemia among pregnant women by survey year.**

for example, women from the poorest quintile were over 3 times as likely to have anemia (OR = 3.2; 95% CI: 2.1–4.7) and women from the poorer quintile were over 2 times as likely to have anemia (OR = 2.4; 95% CI: 1.6–3.6).

Health indicators associated with anemia in simple logistic regression analysis included pregnancy duration, BMI, number of prior births, tobacco use, and barriers to healthcare. Pregnant women were more likely to have anemia if they were in their 2nd trimester (OR = 2.5; 95% CI: 1.8–3.4) or their 3rd trimester (OR = 1.5; 95% CI: 1.1–2.1) compared to pregnant women in their 1st trimester. A normal BMI was also associated with anemia during pregnancy compared to a BMI indicating that a pregnant woman was overweight or obese (OR = 1.5; 95% CI: 1.1–1.9). Compared to not having had any prior births, pregnant women with 3 or more prior births were more likely to have anemia (1.5; 95% CI: 1.0–2.1). Pregnant women using tobacco were more likely to have anemia compared to pregnant women not using tobacco (2.1; 95% CI: 1.2–3.5). Finally, reporting 1 or more barriers to accessing healthcare was associated with anemia during pregnancy (1.4; 95% CI: 1.1–1.8).

Geographic regions of residence were likewise associated with a woman having anemia during pregnancy. Pregnant women from rural areas were more likely to have anemia compared to pregnant women from urban areas (1.7; 95% CI: 1.3–2.2). Likewise, pregnant women living outside of the area surrounding Phnom Penh were more likely to have anemia compared to pregnant women from the Phnom Penh area (1.6; 95% CI: 1.1–2.3). Coastal regions were

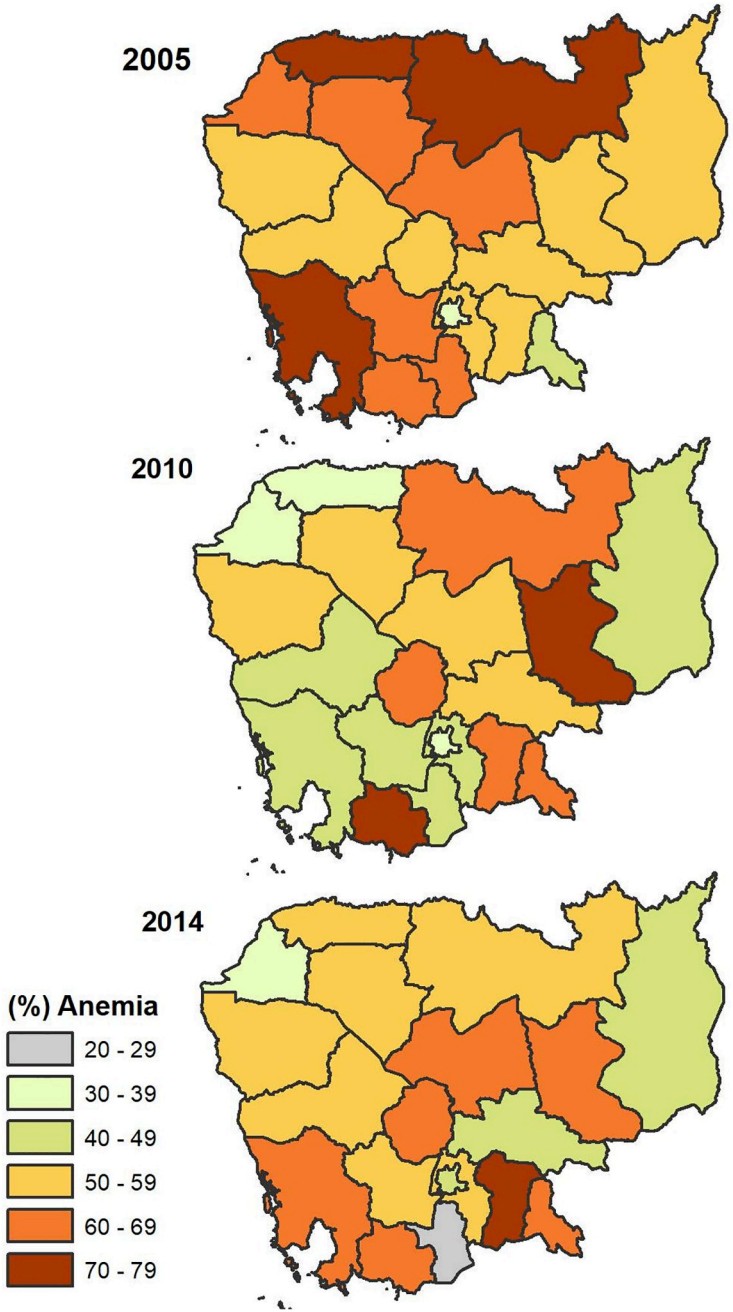

**Fig 3. Geographical distribution of anemia among pregnant women aged 15–49 years old.** Map created using ArcGIS software version 10.8 [26]. Shapefiles for administrative boundaries in Cambodia are publicly accessible through DHS website at [https://spatialdata.dhsprogram.com/boundaries/#view=table&countryId=KH] [27]. CDHS data are publicly accessible through the DHS website at (https://dhsprogram.com/data/available-datasets.cfm) [28].

positively associated with anemia compared to plains regions (1.9; 95% CI: 1.2–2.9) as were mountain regions (1.4; 95% CI: 1.0–2.0).

In the final multiple logistic model (Fig 5), factors associated with anemia included age, wealth index, pregnancy duration, BMI, and geographical regions. Pregnant women were

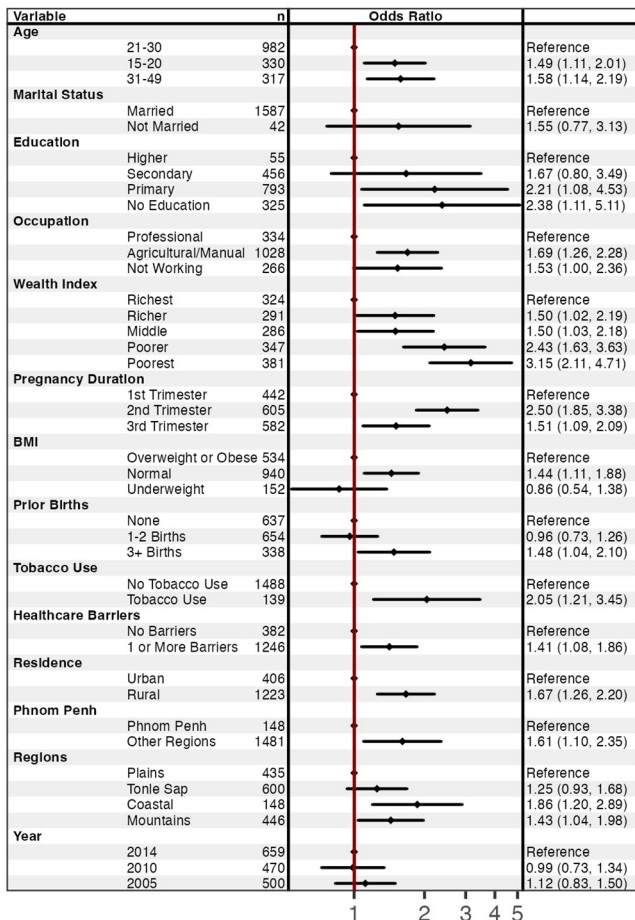

**Fig 4. Factors associated with anemia among pregnant women aged 15–49 years old in simple logistic regression analysis (n = 1,567).** Forest plots were generated using the *forest model* package in R.

more likely to have anemia if they were aged 31–49 (AOR = 1.6; 95% CI: 1.0–2.4) compared to pregnant women aged 21–30. Compared to pregnant women from the richest wealth quintile, pregnant women from the poorest quintile were almost 3 times as likely to have anemia (AOR = 2.8; 95% CI: 1.6–4.9) as were women from the poorer quintile (AOR = 2.2; 95% CI: 1.3–3.9). Pregnant women were more likely anemic if they were in their 2nd trimester (AOR = 2.6; 95% CI: 1.9–3.6) or 3rd trimester (AOR = 1.6 95% CI: 1.1–2.3) compared to their 1st trimester. Pregnant women with a normal BMI were more likely of developing anemia (AOR = 1.4; 95% CI: 1.0–1.9) compared to pregnant women who were overweight or obese. Pregnant women living in coastal regions had higher odds having anemia (AOR = 1.9; 95% CI: 1.2–3.0) compared to those living in plain region. Statistically significant associations were further visualized as predicted probabilities (see S1 Fig).

## Discussion

Prevalence of anemia among pregnant women in Cambodia decreased slightly from 56% in 2005 to 53% in 2014. Also prevalence higher in children between 2005–2014 [30]. This prevalence level was relatively higher than many Southeast Asian countries during the same time period [31]; for example, in Thailand the prevalence of anemia among pregnant women

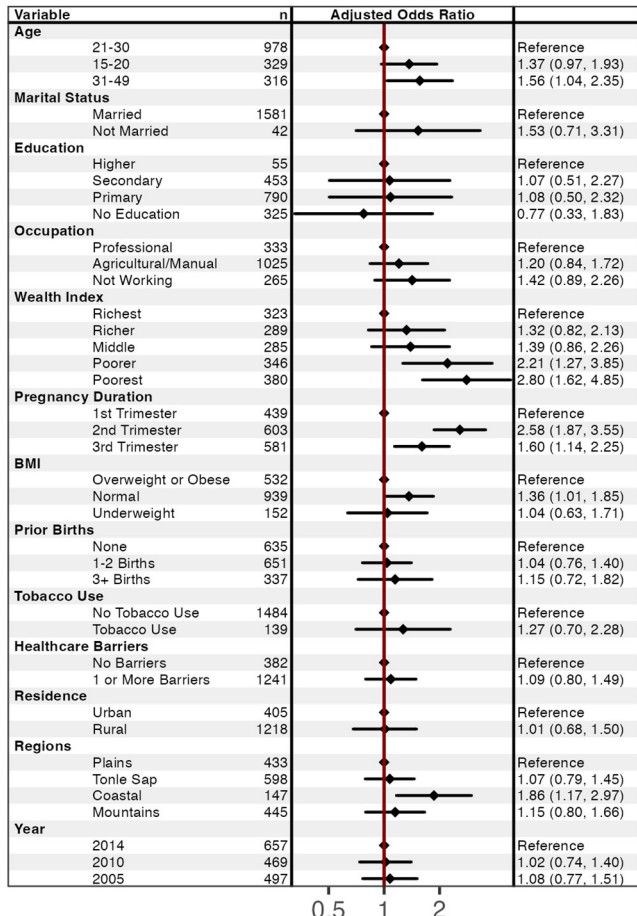

**Fig 5. Factors associated with anemia among pregnant women aged 15–49 years old in multiple logistic regression analysis (n = 1,567).** Forest plots were generated using the forest model package in R.

ranged from 31% in 2005 to 31.9% in 2016 and prevalence of anemia among pregnant women in Vietnam ranged from 33.8% in 2005 to 28.2% in 2015. The elevated prevalence of anemia among pregnant women in Cambodia may be explained by factors related to dietary habits, nutrition deficiencies in micronutrients such as iron or vitamin A, genetic hemoglobin disorders such as the high prevalence of thalassemia in Cambodia (e.g., prevalence rang 50 to 60%) [27, 30], lack of medical treatments, inflammation [32], and malaria infections, and helminths; these potential pathways require further evaluation [16]. Interventions to help mitigate anemia in Cambodia include the promotion of a diversified diet; recommendations that all pregnant women attend at least 4 ANC visit, receive a standard dose of 30–60 mg iron and 400 μg folic acid beginning as soon as possible during gestation, and iron-containing supplements no later than the first trimester of pregnancy; prevention and treatment of malaria; use of insecticide-treated bed nets; helminth prevention and control; delayed cord clamping; and increased birth spacing [33]. However, some have noted that factors addressed by these interventions cannot fully account for the elevated prevalence of anemia [30, 32] and data from the 2000–2014 Cambodia DHS indicated that the percentage of pregnant women attended four or more antenatal care visits has increased dramatically, from 9% in 2000 to 76% in 2014. However proportion

was lowest reported by women resided in rural, plateau, and coastal region, lowest families income, limited education, and first pregnant [9], over 76% took standard iron and folic acid supplements during pregnancy and after delivery through the national health system, while only 65% of pregnant women took iron supplements living in lowest wealth households, which lowest in Preah Vihea/Stung Treng (63.3%) and Mondol Kiri/Ratanakiri (56%) [5]. In the last 10 years, Cambodia has also piloted weekly iron and folic acid supplementation for women of reproductive age. However, despite evidence for an impact on iron status during pregnancy, factors such as cost and need for behavioral change have hampered implementing this policy on a large scale [5]. In contrast, food fortification efforts, such as iron-fortified fish sauce, are currently underway [34] and 72% of women received medications to combat parasites [5]; hence the interest in better understanding associations between anemia and demographic, economic, and social factors as well as health-seeking behaviors across different parts of the country [30, 35].

Our results suggest associations between anemia in pregnant women and age, wealth, pregnancy duration, BMI, and region of residence that are robust to adjusting for other factors. Older aged pregnant women were more likely to have anemia compared to young pregnant women, which is consistent with previous studies [11–13] and may reflect biological and physical disadvantages with increased maternal age; for example, developing gestational diabetes, high blood pressure during pregnancy, a mother having multiple pregnancies, and enduring the collective impact of exhausting labor-related complications [36–38]. Consistent with studies from Rwanda [39], Ethiopia [40], and Uganda [41], pregnant women were also more likely to have anemia if they were from the poorest wealth quintiles with the patterns of association between anemia and wealth exhibiting a dose response to increased poverty. Lower socioeconomic status is often associated with negative consequences regarding consumption of a healthy diet, greater chance of infection diseases, and less access to healthcare services [42, 43]. In contrast, individuals from higher socioeconomic groups have a comparative advantage that enables them to purchase sufficient food with greater variety and quality, thus protecting against anemia [44]. Not only pregnant women, but also children from wealthy families, were less likely to suffer anemia than children from poorer families [30]. In addition, pregnant women living in the coastal regions of residence were more likely have anemia compared to women living in plains regions. These differences in the prevalence of anemia might be due to variations in socioeconomic status, attention given to focused antenatal care and supplementation of iron sulfate throughout the pregnancy, dietary patterns, sample size, and geographical and lifestyle variations [45]. Finally, pregnant women with a normal BMI were more likely of have anemia compared to pregnant women who were overweight or obese, which is surprising given prior studies suggesting that higher BMI is negatively associated with iron status among pregnant women [46]. This association may be related to nutrition; e.g., it may be that overweight or obese women have a higher iron intake compared to normal weigh women [34]. However, the level of iron content of foods consumed by pregnant women in Cambodia remains unclear, hence future research should employ a nutrition survey for pregnant women and evaluate these potential associations directly.

This study was conducted using pooled Demographic and Health surveys for 2005, 2010, and 2014, which enabled us to perform a descriptive analysis of temporal and geographical trends in anemia among pregnant women and also analyze potential risk factors using nationally representative data and generalize our findings to pregnant women throughout Cambodia. However, using cross-sectional data limits our ability to assess causal relationship among the associated factors that we identified. Using secondary data that was gathered to collect health and demographic data across a broad variety of variables related to child and maternal health enabled us to evaluate some of the factors likely associated with anemia among pregnant

women, but other factors such as diet, nutritional supplements (Vitamin B12, folate); malaria infection, non-communicable diseases such as hypertension and diabetes, and awareness about anemia were excluded from the analysis due to data limitations. Focusing exclusively on pregnant women yielded a relatively smaller sample size; some categorical variables were combined into larger categories for inclusion in our analysis, which resulted in some loss of information in these variables. Finally, the most recent round of CDHS data that includes anemia indicators was gathered in 2014, which may no longer reflect present prevalence of anemia among pregnant women in Cambodia. Future research should further evaluate these relationships and the general prevalence of anemia among pregnant Cambodian women once additional data becomes available (unfortunately, Hemoglobin measurements were not collected in CDHS 2021–22) [47]. With additional data, future research should consider a formal spatial analysis for enhancing the scientific rigor of geospatial trends [48].

Our findings suggest that the high prevalence of anemia among pregnant women in Cambodia declined only slightly over the 10-year period between 2005 and 2014. This is concerning given that anemia during pregnancy is a public health problem that should be actively addressed to mitigate the proximal and distal health risks associated with the condition. We identified age, socioeconomic status, higher BMI, and regions of residence as potential risk factors associated with anemia during pregnancy. Public health practitioners and policy makers should consider demographic groups represented by these characteristics for better targeting interventions to address anemia among pregnant women in Cambodia.

## Supporting information

**S1 Fig. Predicted probability of anemia among pregnant women aged 15–49 years old (n = 1,567).** Visualized using "ggplot2" package in R. Probabilities arranged by associated factor; i.e., age, duration of pregnancy, wealth, and region.
(TIFF)

**S1 Table. Prevalence of anemia in the 19 domains in CDHS 2005, CDHS 2010 and 2014.**
(DOCX)

## Acknowledgments

The authors would like to thank the DHS-ICF, who approved the data used for this paper.

## Author Contributions

**Conceptualization:** Samnang Um, Jonathan A. Muir.

**Data curation:** Samnang Um.

**Formal analysis:** Samnang Um, Jonathan A. Muir.

**Investigation:** Samnang Um, Jonathan A. Muir.

**Methodology:** Samnang Um, Jonathan A. Muir.

**Project administration:** Samnang Um.

**Software:** Samnang Um.

**Supervision:** Jonathan A. Muir.

**Validation:** Samnang Um, Jonathan A. Muir.

**Visualization:** Samnang Um, Jonathan A. Muir.

**Writing – original draft:** Samnang Um.

**Writing – review & editing:** Samnang Um, Heng Sopheab, An Yom, Jonathan A. Muir.

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
