## [Decision Letter · Decision Letter 0]

17 Oct 2022

PONE-D-22-24121Trends and factors associated with anemia among pregnant women aged 15-49 years old in Cambodia: Data analysis of the Cambodia Demographic and Health SurveysPLOS ONE

Dear Dr. Um,

Thank you for submitting your manuscript to PLOS ONE. After careful consideration, we feel that it has merit but does not fully meet PLOS ONE’s publication criteria as it currently stands. Therefore, we invite you to submit a revised version of the manuscript that addresses the points raised during the review process.

In addition to the comments of the reviewers, authors should also attend to the attached comments sent by the reviewers. If authors were unable to access the attached comments, the PLOS ONE editorial office should be contacted.  Authors should use number lines for easy referenceIntroduction: second line: change "e.g" to "For example". Do the same throughout the manuscriptThe statistical analysis and results are satisfactory

We look forward to receiving your revised manuscript.

Kind regards,

Gbenga Olorunfemi, MBBS,MSC,FMCOG,FWASC

Academic Editor

PLOS ONE

Journal Requirements:

2. We note that Figure 3 in your submission contain [map/satellite] images which may be copyrighted. All PLOS content is published under the Creative Commons Attribution License (CC BY 4.0), which means that the manuscript, images, and Supporting Information files will be freely available online, and any third party is permitted to access, download, copy, distribute, and use these materials in any way, even commercially, with proper attribution. For these reasons, we cannot publish previously copyrighted maps or satellite images created using proprietary data, such as Google software (Google Maps, Street View, and Earth). For more information, see our copyright guidelines: http://journals.plos.org/plosone/s/licenses-and-copyright.

a. You may seek permission from the original copyright holder of Figure 3 to publish the content specifically under the CC BY 4.0 license.  

Reviewers' comments:

Reviewer's Responses to Questions

**Comments to the Author**

1. Is the manuscript technically sound, and do the data support the conclusions?

Reviewer #1: Partly

Reviewer #2: Partly

2. Has the statistical analysis been performed appropriately and rigorously? 

Reviewer #1: Yes

Reviewer #2: No

3. Have the authors made all data underlying the findings in their manuscript fully available?

Reviewer #1: No

Reviewer #2: Yes

4. Is the manuscript presented in an intelligible fashion and written in standard English?

Reviewer #1: Yes

Reviewer #2: Yes

5. Review Comments to the Author

Reviewer #1: PONE-D-22-24121

Trends and factors associated with anemia among pregnant women aged 15-49 years old in Cambodia: Data analysis of the Cambodia Demographic and Health Surveys

Using three rounds of the Cambodian DHS dataset, authors aimed to evaluate the pattern and contributing factors of anemia among pregnant women. Overall, the manuscript is nicely written, incorporates use of the proper statistical analysis, and draws a conclusion based on the results. Nevertheless, there are a number of issues with this version that must be addressed before it can move further.

Including line number could be helpful to locate the comments

Title

Please revise the title with “Trends and factors associated with anemia among pregnant women in Cambodia: analysis of 2005 to 2014 Cambodia Demographic and Health Surveys”

Abstract

Introduction: Remove the word “roughly 37% (32 million)” and replace with suitable word here.

"Women in developing countries are at higher risk of anemia due to micronutrient deficiencies, hemoglobinopathies, infections, or other socio-demographic factors, especially among pregnant women" this sentence is not justifiable because anemia is highly prevalent among both under five children and women of reproductive age. So, please revise this sentence.

Replace describes with “aimed”

Results: Please also mention the changing pattern of anemia geographically.

Manuscript

Introduction:

Replace g/dl with “g/dL”

"In 2019, it was estimated that anemia affected approximately 37% (32 million) of pregnant women worldwide…….." you can catch this information here: https://www.thelancet.com/journals/langlo/article/PIIS2214-109X(22)00084-5/fulltext#:~:text=Globally%2C%20the%20prevalence%20of%20anaemia,%25%20(34%E2%80%9339).

Reference 4 do not have the mention information. Please insert the suitable sources here. You can get this information in above mentioned link.

Is reference 9 reliable source?

Given that other research have already been published utilizing the CDHS dataset, the authors should discuss other studies that used the same data and explain what makes their study unique. Most likely demonstrating trend analysis, geographic variation, and the significance of each. Furthermore, the study did not break out the variables affecting anemia by geographical area. The report only lists the geographic regions where anemia is most common. Given that both studies use the same data, the author needs to explain why this one is more significant than the others indicated above.

Materials and methods

Fig 1 do not provide complete information regarding the sample size selection process. How did you select the (sub) sample? What are the inclusion/exclusion criteria? Detail information is required to make clearer.

It would be preferable if you included all of these coding strategies in a table and uploaded it as a supporting document. What matters most in the manuscript is how you recategorized the predictor variables.

Not multivariate, use “multivariable logistic regression” throughout the manuscript.

Authors have mentioned "A significant level of any covariates at p-value < 0.15 were included in the multivariate logistic regression analysis" why did you include only <0.15 value? Most of the papers have mentioned <0.2.

The authors reported on the multicollinearity check, but they failed to indicate the cutoff value that was employed in this study. I could not locate the report of multicollinearity as well.

Results

Instead of pooled data, Table 1 should display data from all three DHS survey rounds. It is logical. Therefore, please revise it accordingly.

Discussion

The authors have mentioned that "The elevated prevalence of anemia amongbpregnant women in Cambodia may be explained by dietary habits, deficiencies in micronutrients suchb as iron or vitamin A, hemoglobinopathy, malaria infections, and helminths" that means does dietary pattern is different from the neighboring countries? Please explain it in detail.

The mentioned programs are the public health intervention implemented in Cambodia. It would be better if you mention the compliance and effectiveness of such program to mitigate the anemia among pregnant women in Cambodia.

Age is still contradictory. Some studies concluded that getting older is protective, while others countered that it makes women more susceptible to anemia. Please discuss.

Some studies are:

https://bmjopen.bmj.com/content/11/3/e041982.long#ref-39

https://bmjopen.bmj.com/content/bmjopen/9/Suppl_3/1.full.pdf
https://pubmed.ncbi.nlm.nih.gov/23516270/

Why pregnant women having normal BMI had higher odds of anemia compared to overweight/obese women?

Disparities of anemia according to geographical pattern might be explained here: https://bmjopen.bmj.com/content/11/3/e041982.long#ref-64

Reviewer #2: Statistical analysis

The authors used logistic models to determine factors associated with anemia among pregnant women aged 15-49 years old in Cambodia. They went on indicating that they fitted bivariate and multivariate logistic regression models. This is confusing as only one response variable was considered. The concepts of bivariate and multivariate are commonly used when the objective of modelling is to jointly analyse two or outcome variables. Thus, I suggest they change to simple and multiple logistic regression models instead of bivariate and multivariate logistic regression models, respectively.

The analysis aimed at describing the temporal and geographic trends of anemia in pregnant. In this paper, the authors have just provided a description of the temporal and geographic trends by merely using a chart and maps of prevalence of anemia. It is not evident that these spatio-temporal trends described in this paper are significant or not. I would suggest they consider using spatio-temporal modelling approach to this end. Furthermore, they need to extend the time window as three data points (3 years) are too little for a temporal trend analysis.

Finally, the analysis is lacking the model diagnostic that ensures that final model is a good fit. This section may include some of the following: Deviance, R-squared for logistic regression, Likelihood ratio-test, Omnibus Test of model coefficients, Hosmer-Lemshow Goodness of fit test, Classification tables & ORC, Wald test, Analysis of residuals, and Validation of Results (Use the validation sample (test) to assess the external validity and practical significance of the model).

The conclusions drawn can be improved once the above issued are considered. This manuscript is presented in an intelligible style and is written in standard English though there are few typos as indicated n the main document.

In addition to availability of the data, the authors should avail the codes (R) used.

6. PLOS authors have the option to publish the peer review history of their article (what does this mean?). If published, this will include your full peer review and any attached files.

Reviewer #1: No

Reviewer #2: No

---

## [Author Response · Author response to Decision Letter 0]

26 Jan 2023

Responses to the reviewers' feedback were provided in a separate document uploaded with the revised manuscript.

---

## [Decision Letter · Decision Letter 1]

29 Mar 2023

PONE-D-22-24121R1Trends and factors associated with anemia among pregnant women in Cambodia: Analysis of 2005 to 2014 Cambodia Demographic and Health SurveysPLOS ONE

Dear Dr. Um,

Thank you for submitting your manuscript to PLOS ONE. After careful consideration, we feel that it has merit but does not fully meet PLOS ONE’s publication criteria as it currently stands. Therefore, we invite you to submit a revised version of the manuscript that addresses the points raised during the review process.

We look forward to receiving your revised manuscript.

Kind regards,

Gbenga Olorunfemi, MBBS,MSC,FMCOG,FWASC

Academic Editor

PLOS ONE

Journal Requirements:

Reviewers' comments:

Reviewer's Responses to Questions

**Comments to the Author**

1. If the authors have adequately addressed your comments raised in a previous round of review and you feel that this manuscript is now acceptable for publication, you may indicate that here to bypass the “Comments to the Author” section, enter your conflict of interest statement in the “Confidential to Editor” section, and submit your "Accept" recommendation.

Reviewer #1: (No Response)

2. Is the manuscript technically sound, and do the data support the conclusions?

Reviewer #1: Yes

3. Has the statistical analysis been performed appropriately and rigorously? 

Reviewer #1: I Don't Know

4. Have the authors made all data underlying the findings in their manuscript fully available?

Reviewer #1: No

5. Is the manuscript presented in an intelligible fashion and written in standard English?

Reviewer #1: Yes

6. Review Comments to the Author

Reviewer #1: Manuscript ID: PONE-D-22-24121R1

Title: Trends and factors associated with anemia among pregnant women in Cambodia: Analysis of 2005 to 2014 Cambodia Demographic and Health Surveys

The most of the comments have been addressed by the authors; nevertheless, a few remarkable issues remain that should be addressed before a decision is made.

General comments

1.Also, I couldn't find information about the construction of forest plots for Figs 4 and 5. Please provide information in the manuscript about how forest plots were constructed.

3. Authors have claimed to evaluate temporal and geographical trends in anemia among pregnant women using nationally representative data in Cambodia. However, I haven't noticed the findings of geographical trends in this study. It needs robust spatial analysis.

4. To display the descriptive results at different times, I still strongly recommend presenting the descriptive data for all three rounds of surveys in Table 1.

Specific comments

1. The X axis label in Fig 2 needs to be corrected to a percentage, which looks confusing.

2. Discussion: In Cambodia, despite the implementation of various public health programs, the prevalence of anemia among pregnant women remains stagnant. Please discuss plausible reason and ways to mitigate the problem.

3. Why pregnant women with a normal BMI were more likely to have anemia compared to pregnant women who were overweight or obese? Please explain the plausible physiological mechanism.

4. I had recommended including the evidence of disparities of anemia according to geographical pattern in the previous round, but I could not locate it.

https://bmjopen.bmj.com/content/11/3/e041982

7. PLOS authors have the option to publish the peer review history of their article (what does this mean?). If published, this will include your full peer review and any attached files.

Reviewer #1: No

---

## [Author Response · Author response to Decision Letter 1]

27 May 2023

2nd revision has been followed reviewer sugguested inhiligted

---

## [Decision Letter · Decision Letter 2]

25 Sep 2023

PONE-D-22-24121R2Anemia among Pregnant Women in Cambodia: A Descriptive Analysis of Temporal and Geospatial Trends and Logistic Regression-Based Examination of Factors Associated with Anemia in Pregnant WomenPLOS ONE

Dear Dr. Um,

Thank you for submitting your manuscript to PLOS ONE. After careful consideration, we feel that it has merit but does not fully meet PLOS ONE’s publication criteria as it currently stands. Therefore, we invite you to submit a revised version of the manuscript that addresses the points raised during the review process.

We look forward to receiving your revised manuscript.

Kind regards,

Germana Bancone, Ph.D

Academic Editor

PLOS ONE

**Additional Editor Comments:**

Together with the new requests from the reviewers, these requests from previous reviews have not been addressed in the current version and should be in the next revision.

1.I am not convincing with the using the "unadjusted/adjusted" instead of bivariate/multivariable. As suggested by the reviewer in the previous round, I would rather recommend to use simple and multiple logistic regression models. Also, I couldn't find information about the construction of forest plots for Figs 4 and 5. Please provide information in the manuscript about how forest plots were constructed.

The authors provide details on how the Forest plot was created in R in the legend of the figure but this should instead be detailed in the Methods section of the paper.

2. I had asked this query in the previous round "A significant level of any covariates at p-value < 0.15 were included in the multivariate logistic regression analysis" why did you include only <0.15 value? Most of the papers have used <0.2. The author claims the p-value of 0.15 is relevant from a theoretical standpoint. It is not clear. Please clarify it in simple and understandable way.

Reviewers' comments:

Reviewer's Responses to Questions

**Comments to the Author**

1. If the authors have adequately addressed your comments raised in a previous round of review and you feel that this manuscript is now acceptable for publication, you may indicate that here to bypass the “Comments to the Author” section, enter your conflict of interest statement in the “Confidential to Editor” section, and submit your "Accept" recommendation.

Reviewer #3: (No Response)

2. Is the manuscript technically sound, and do the data support the conclusions?

Reviewer #3: Partly

3. Has the statistical analysis been performed appropriately and rigorously? 

Reviewer #3: No

4. Have the authors made all data underlying the findings in their manuscript fully available?

Reviewer #3: Yes

5. Is the manuscript presented in an intelligible fashion and written in standard English?

Reviewer #3: Yes

6. Review Comments to the Author

Reviewer #3: Methods

1. The data should be better explained to give the reader confidence about the categorization of some of the variables. For example, less than 3 previous births as the reference category – what about women without a previous pregnancy? It is not clear whether these are included in the reference category (and why this would be reasonable to do).

2. Some concerns around the inclusion of variables in the adjusted model if their association with the outcome was associated with p-value <= 0.15.

The authors should include all covariates in the adjusted model which are known risk factors for anaemia. Results presented for the regression analysis seem to consider all covariates in the adjusted model despite the authors stating only covariates deemed significant from the univariate analysis would be considered. Further justification for the inclusion of covariates in the model should be provided.

3. In the analytical plan, “complex survey design” was accounted for in the regression analysis but it is unclear what this means. The authors should provide further explanation.

Results

4. In the results section, it would be clearer to provide the percent (proportion) when reporting summary statistics from Table 1. Otherwise, the reader has to check these proportions themselves.

5. Figure resolution needs to be improved on all figures.

Discussion

6. Reference to paper on childhood anemia by the authors and a comparison between the results of this study and the childhood anaemia study would be helpful for conceptualizing this research.

7. Authors should provide more details on the limitations of the analysis - particuarly on the categorization of some of the variables. This would give the reader more confidence that these limitations were considered in the interpretation of results.

7. PLOS authors have the option to publish the peer review history of their article (what does this mean?). If published, this will include your full peer review and any attached files.

Reviewer #3: No

---

## [Author Response · Author response to Decision Letter 2]

9 Nov 2023

Responses to reviewer and editor comments are provided with an uploaded document

---

## [Editor Report · Decision Letter 3]

21 Nov 2023

Anemia among Pregnant Women in Cambodia: A Descriptive Analysis of Temporal and Geospatial Trends and Logistic Regression-Based Examination of Factors Associated with Anemia in Pregnant Women

PONE-D-22-24121R3

Dear Dr. Um,

We’re pleased to inform you that your manuscript has been judged scientifically suitable for publication and will be formally accepted for publication once it meets all outstanding technical requirements.

Kind regards,

Germana Bancone, Ph.D

Academic Editor

PLOS ONE
---

## [Editor Report · Acceptance letter]

23 Nov 2023

PONE-D-22-24121R3 

Anemia among Pregnant Women in Cambodia: A Descriptive Analysis of Temporal and Geospatial Trends and Logistic Regression-Based Examination of Factors Associated with Anemia in Pregnant Women 

Dear Dr. Um:

I'm pleased to inform you that your manuscript has been deemed suitable for publication in PLOS ONE. Congratulations! Your manuscript is now with our production department. 

Kind regards, 

on behalf of

Dr. Germana Bancone 

Academic Editor

PLOS ONE